# Experimental Investigation on the Effect of Coil Shape on Planar Eddy Current Sensor Characteristic for Blade Tip Clearance

**DOI:** 10.3390/s24186133

**Published:** 2024-09-23

**Authors:** Lingqiang Zhao, Yaguo Lyu, Fulin Liu, Zhenxia Liu, Ziyu Zhao

**Affiliations:** 1School of Power and Energy, Northwestern Polytechnical University, Youyi West Road 127#, Xi’an 710054, China; lqznpu@mail.nwpu.edu.cn (L.Z.); fulinliu@mail.nwpu.edu.cn (F.L.); zxliu@nwpu.edu.cn (Z.L.); 2Xi’an Research Institute of High-Tech, Xi’an 710054, China; zziyu@mail.nwpu.edu.cn

**Keywords:** planar eddy current sensor, coil shape, blade tip clearance, inductance, measurement range, sensitivity, linearity

## Abstract

Given the increasing application of eddy current sensors for measuring turbine tip clearance in aero engines, enhancing the performance of these sensors is essential for improving measurement accuracy. This study investigates the influence of coil shape on the measurement performance of planar eddy current sensors and identifies an optimal coil shape to enhance sensing capabilities. To achieve this, various coil shapes—specifically circular, square, rectangular wave, and triangular wave—were designed and fabricated, featuring different numbers of turns for the experiment at room temperature. By employing a method for calculating coil inductance, the performance of each sensor was evaluated based on key metrics: measurement range, sensitivity, and linearity. Experimental results reveal that the square coil configuration outperforms other shapes in overall measurement performance. Notably, the square coil demonstrated a measurement range of 0 mm to 8 mm, a sensitivity of 0.115685 μH/mm, and an impressive linearity of 98.41% within the range of 0 mm to 2 mm. These findings indicate that the square coil configuration enhances measurement capabilities. The conclusions drawn from this study provide valuable insights for selecting coil shapes and optimizing the performance of planar eddy current sensors, thereby contributing to the advancement of turbine tip clearance measurement techniques in aero engines.

## 1. Introduction

The blade tip clearance of a gas turbine has a significant impact on its performance and efficiency. Therefore, the precise measurement of blade tip clearance is of great importance for its accurate design and optimization [1,2,3,4]. The study of high precision and large measurement ranges for tip clearance is necessary and crucial. Various non-contact gap measurement technologies have been developed and applied, including capacitive, optic, microwave, and eddy current [5,6,7,8,9,10,11,12].

Due to the advantages of eddy current sensors, such as non-contact operation, high-temperature resistance, high reliability, fast response time, strong anti-interference capabilities, and the ability to work in harsh environments, it has been widely researched and applied in the measurement of blade tip clearance at the turbine section of aircraft engines [10,11,12]. Since 2013, researchers from Oxford University, led by K.S. Chana, developed eddy current sensors capable of withstanding temperatures exceeding 1500 K. In addition, performance tests were conducted on the RB168Mk 101 Spay engine (made by Rolls-RoyceGroupPlc)and Rolls-Royce VI-PER engine(made by Rolls-RoyceGroupPlc) [13,14,15]. The test result not only showed that the eddy current sensor (ECS) is feasible for the measurement of blade tip clearance, but also that the results are reliable. Additionally, based on eddy current technology, K.S. Chana and V. Sridhar developed the Blade Tip Timing system which utilizes optimized algorithms to build simple, reliable, real-time, and cost-effective analog electronic circuits [16]. J Zhe et al. have simplified the calibration process of eddy current sensors by introducing new mutual inductance calculation methods [17,18,19]. They have not only confirmed that the planar coil possesses higher sensitivity in tip clearance measurement, but also verified the eddy current method in the laboratory with 3000 r/min revolution and 1300 K temperature. Zhao Z.Z et al. [20,21,22] focused on the design and validation of the high-temperature planar eddy current sensor. The simulation and experiment methods were both used to determine the optimal sensor structure parameters and materials. The designed sensor was tested to prove that it can operate normally at extremely high temperatures. 

Based on the above, the eddy current method presents a promising approach for monitoring the dynamic blade tip clearance in turbines. Through extensive experimentation, researchers have explored the performance of eddy current sensors under extreme high-temperature conditions. For tip clearance sensors, it is essential to ensure not only high-temperature resistance but also a large linear range and high sensitivity. The sensitivity and linear range of eddy current sensors are primarily influenced by the magnetic field distribution of the coil [23], which is directly affected by the coil’s shape and indirectly by the measured target [24]. Therefore, studying the impact of coil shape on sensor performance is of practical significance. Understanding these influences can further advance the application of eddy current sensors in measuring tip clearance in aero engines, potentially leading to more accurate and reliable measurements in challenging operational environments.

Building on previous research, this paper focuses on the application of eddy current sensors for measuring turbine tip clearance. The study utilizes platinum, renowned for its excellent high-temperature performance, to fabricate coils with various shapes and turns, including rectangular, circular, triangular, square, and square-wave inductive coils. For simplification, the curved turbine blade is represented as a straight blade in this research. A dynamic rotor experimental platform is set up, and a novel method for calculating sensor inductance during dynamic measurements is proposed. The study explores and analyzes the effects of different effective blade sections, excitation frequencies, and excitation voltages on the sensor’s characteristics. Based on this analysis, the optimal effective section, excitation frequency, and excitation voltage are determined, allowing for a detailed examination of the impact of various coil shapes on the sensor’s measurement range, sensitivity, and linear range. This research aims to support the optimized design of eddy current turbine tip clearance sensors to enhance their performance. The findings are expected to contribute significantly to the development of more accurate and reliable sensors for aero-engine applications.

## 2. Design Concept

### 2.1. The Basic Principle of the Eddy Current Sensor

The principle underlying eddy current gap measurement is based on Faraday’s law of electromagnetic induction and Lenz’s law. In this process, an ECS is energized by a high-frequency alternating current (AC) signal, which creates a primary alternating magnetic field. When a moving metallic blade traverses this magnetic field, eddy currents are induced within the blade. These induced currents produce a secondary alternating magnetic field that opposes the primary one, as described by Lenz’s law. This interaction results in changes in various parameters of the coil, which the ECS is designed to detect and measure. The alteration in the magnetic field due to the presence and movement of the metallic blade enables precise gap measurements. A diagram illustrating this process can be found in Figure 1, which visually represents the interaction between the primary and secondary magnetic fields and the induced currents in the blade.

Figure 1a illustrates an eddy current sensor mounted on the turbine case, including a disk of blades. As shown in Figure 1b, the sensor can be modeled as an inductance (L_c_) connected with a resistance (R_c_) in series; the blade is modeled in the same way. The sensing mechanism is as follows: the sensing coil which is excited by high frequency AC signal generates a primary alternating magnetic field. When a blade passes the coil, it induces an eddy current inside the blade, which generates a secondary alternating magnetic field which is contrary to the primary alternating magnetic field, ultimately resulting in a decrease in the coil inductance L_c_ [25]. It shows that the smaller the tip clearance, the larger the L_eq_ change in Figure 1c. Moreover, based on the equivalent circuit diagram, the equation for calculating the equivalent inductance (L_eq_) of the coil is as follows [21].
(1)$$L_{eq}=L_{c}-\frac{w^{2}M^{2}}{R_{b}^{2}+w^{2}L_{b}^{2}}L_{b}$$
wherein, ω is angular frequency; M is mutual inductance coefficient between the coil and blade.

Equation (1) shows that the equivalent inductance (L_eq_) is not only related to the inductance (L_c_ and L_b_) and resistance (R_c_ and R_b_) of the coil and the blade but also to the mutual inductance coefficient (M) between them and the excitation frequency (ω). According to the literature [26,27], the M value is related to the number of turns (N), the shape, and permeability of the coil, as well as the relative position and shape of the blade.

The equivalent inductance (*L_eq_*) of the coil is related to the effective cross-sectional area facing the target [28]. The effective cross-sectional area is explained in Figure 2. The square wave shape coil is used, but the coil type can be of any shape. The effective area of the coil changes in accordance with the position of the blade, which affects the coupling between the coil and the target.

When the coil shape is established, the output parameters of the sensing coil are influenced by several factors, including the excitation signal characteristics, the number of coil turns, and the installation position. Therefore, this paper will conduct a comprehensive study on how coil shape affects sensor output performance from several aspects, including excitation signal voltage, frequency, number of coil turns, and the relative position between the sensing coil and the blade. Through this multidimensional analysis, the study aims to provide actionable insights for optimizing eddy current sensor designs, enhancing their performance in gap measurement applications.

### 2.2. Design of Various Shapes of Sensing Coil

Each type of sensing coil has the same external dimensions (10 mm × 10 mm), which is wound with platinum wire of 0.2 mm in diameter. Figure 3 shows that the various sensing coils are categorized into six main types, totaling 18 different shapes of coils. The gap between the coils is uniformly distributed. As long as the coils are within the fixed external area, the gap size is determined by the number of turns of the coil.

Type G, H, I, J, K, and L are of the rectangular wave coil with both vertical and horizontal position settings with different coil turn numbers. Type G, H, and I are positioned vertically to the blade’s chord direction, as shown in Figure 2. But Types J, K, and L are positioned horizontally. Type G and J have 10 coil turns, respectively. Type H and K have seven coil turns, respectively. Type F and L have five coil turns, respectively. Type L, M, N, O, P, Q, and R are of the triangular wave coil with both vertical and horizontal position settings with different coil turn numbers. Type M, N, and O are positioned vertically to the blade’s chord direction. Furthermore, Types P, Q, and R are positioned horizontally. Type M and P have 10 coil turns, respectively. Type N and Q have seven coil turns, respectively. Type O and S have five coil turns, respectively. Type A, B, and C are of the circular coil with 10, 7, and 5 turns, respectively; Type D, E, and F are of the square coil with 10, 7, and 5 turns, respectively.

### 2.3. Fabrication of the Sensor

Currently, the production of the coil using platinum wire is not well-developed. In order to create coils of various shapes, the following coil fabrication tools were made, as shown in Figure 4. The required coil models in various shapes are carved by a Computer numerical control machine (CNC), and the groove width is 0.3 mm on an aluminum plate, as shown in Figure 4. Then the grooves of the entire model are filled with platinum wire of 0.2 mm in diameter. Note that a small amount of epoxy was applied on the surface of the platinum wire to retain the shape of the spiral coil during wire winding. After the sensing coil was formed, the sensing coil was carefully removed and attached to a ceramic plate with dimensions of 10 mm × 10 mm and a thickness of 1 mm, as followed in Figure 5. Then assemble the ceramic plate, sensing coil, and acrylic casing and fill the ceramic materials (Ceramacast 645N, Aremco, in USA). The completed sensor is shown in Figure 6.

### 2.4. Experimental Setup

To avoid the impact of blade elongation due to centrifugal force on sensor performance, a specialized testing setup was designed and manufactured, as illustrated in Figure 7a. In this configuration, the sensor is mounted vertically on a pre-balanced disc, which is positioned horizontally and connected to a brushless motor (80BL01 220 V brushless motor, made by Okoda stepper motor in China) via a stainless-steel shaft. The rotational speed of the disc can be adjusted between 1000 r/min and 18,000 r/min, with precise measurements taken using an optical tachometer (LG9200 made by Xiaoye Measurement Technology). Note that all the experiments were conducted at a constant speed of 2500 r/min at room temperature in this paper. The sensor is securely fastened using a fixture attached to a 3axis high precision stage. This design ensures that the sensor remains on the horizontally positioned disc, effectively eliminating the impact of blade elongation caused by centrifugal forces on the sensor output. In this setup, the tip clearance is defined as the distance from the underside of the sensor to the top surface of the blade. The clearance between the sensor and the blade is meticulously controlled by the 3-axis high precision stage, which offers an adjustment precision of 0.5 μm along the *Z*-axis and 10 μm along the X and Y axes. This level of precision allows for accurate measurement of the gap, ensuring reliable sensor performance under varying operational conditions.

## 3. Testing, Result, Discussion

### 3.1. Tip Clearance Measurement and Signal Processing

The sensing coil *L_c_* can reflect the tip clearance. To measure *L_c_*, an AC sin wave excitation was applied by a function generator (Keysight 33600A made by Keysight in USA). The sensing coil was connected in series with a resistor (*R*_0_ = 10 Ω), as shown in Figure 8a; the output signals, *V*_1_ and *V*_2_, were recorded by a digitizer (4347, GaGe made by GAGE in USA) at a sampling rate of 100 MHz in all experiments.

To accurately calculate the inductance value of each sensing coil by MATLAB, first, a Butterworth filter was applied to process the original signals of *V*_1_ and *V*_2_; next, the processed signals of *V*_1_ and *V*_2_ were divided into many segments of data; each segment consists of 1 μs data. Cubic spline interpolation was then applied to each segment of *V*_1_ and *V*_2_ to reduce the digitization errors caused by the data acquisition. Peak values of individual voltage components and the phase difference (ϴ, as shown in Figure 8b) between the two signals of *V*_1_ and *V*_2_ were obtained by using a peak detection code written in Matlab^®^. Last, from the output voltages (*V*_1_ and *V*_2_) and phase difference (*ϴ*), the *L_c_* of the sensing coil can be calculated by:(2)$$I=\left|\frac{\vec{V_{1}}-\vec{V_{2}}}{R_{0}}\right|=\frac{\sqrt{V_{1}^{2}-2V_{1}V_{2}\cos\theta +V_{2}^{2}}}{R_{0}}$$
(3)$$\alpha =\arcsin\frac{V_{2}\sin\theta }{IR_{0}}=\arcsin\frac{V_{2}\sin\theta }{\sqrt{V_{1}^{2}-2V_{1}V_{2}\cos\theta +V_{2}^{2}}}$$
(4)$$V_{Lc}=V_{2}\sin(\alpha +\theta )=V_{2}\sin(\arcsin\frac{V_{2}\sin\theta }{\sqrt{V_{1}^{2}-2V_{1}V_{2}\cos\theta +V_{2}^{2}}}+\theta )$$
(5)$$L_{c}=\frac{V_{Lc}}{2\pi fI}=\frac{V_{2}R_{0}\sin\alpha }{2\pi f\sqrt{V_{1}^{2}-2V_{1}V_{2}\cos\theta +V_{2}^{2}}}$$
where *I* is the alternating current in the series circuit; *V_Lc_* is the voltage across the *L_c_* in the sensing coil, as shown in Figure 8c; *f* is the excitation frequency.

### 3.2. Result

#### 3.2.1. Effect of Effective Area

As can be seen from Equation (1) in Section 2.1, the inductance of the coil is related to the effective area facing it. Therefore, taking into account the chordwise shape of the blade, the average widths at three positions (as shown in Figure 9) along the blade chord are selected as the blade widths for the blades in the disc, as shown in the Figure 9, and the average widths are 12 mm, 10 mm, and 8 mm.

Figure 10 shows the effect of effective area on the inductance of the circular shape coil (Type A as shown in Figure 3a) at three different areas, and the voltage and frequency of the excitation signal are 1 Vpp and 1 MHz. The inductance value increases as the distance between the sensor and blade increases, regardless of the effective area. However, the inductance value of the coil increases at the same distance as the effective area varies from position 1 to position 3, such as when the distance is 0 mm, the inductance value at position 1 is 0.67425 μH, the inductance value at position 3 has increased to 0.77515 μH. The increase in inductance values is due to a reduction in effective area, which causes a decrease in the mutual inductance coefficient, similar to how an increase in distance leads to an increase in inductance values. As the distance increases to around 8 mm, the inductance value gradually stabilizes. At position 1, the measurement range is from 0 mm to 8.5 mm. But it is from 0–7 mm at the position 3. The sensor sensitivity varies directly proportional to the effective area. Since the variation in turbine blade tip clearance typically ranges from 1–2 mm, our primary focus is on the sensitivity of the sensor within this range. The max sensitivity for three positions, within the range of 0–2 mm is 0.11488 μH/mm, 0.11384 μH/mm, and 0.08229 μH/mm, respectively.

#### 3.2.2. Effect of Excitation Frequency

Figure 11a,b show the effect of excitation frequency on the inductance of the circular coil and square coil (Type A and Type C as shown in Figure 3a) at position 1 (as mentioned in Section 3.2.1), and the excitation signal has a voltage of 1 Vpp, with a varying frequency from 1 MHz to 4 MHz. Since measuring the turbine blade tip clearance requires a high excitation frequency for the sensor, the investigation primarily focused on excitation frequencies above 1 MHz. For both circular and square coils, the inductance values at the starting position of 0 mm decreases as the frequency increases. It can be observed that the inductance values of the circular and square coils have significantly by 0.04334 μH and 0.03736 μH, respectively. These decreases, respectively, account for 12.12% and 13.15% of the inductance change values for the circular coil and the square coil. This significant decrease in inductance values is attributed to the iron losses in the blade due to the eddy current effect. 

The sensor sensitivity varies directly proportional to the excitation frequency. It can be seen that the max sensitivity of the circular coil has increased from 0.11488 μH/mm at 1 MHz to 0.116795 μH/mm at 4 MHz, within the range of 0–2 mm. Similarly, the max sensitivity of the square coil has increased from 0.11185 μH/mm at 1 MHz to 0.115685 μH/mm at 4 MHz.

#### 3.2.3. Effect of Excitation Voltage

Figure 12a,b show the effect of excitation voltage on the inductance of the circular coil and square coil (Type A and Type C as shown in Figure 3a) at position 1 (as mentioned in Section 3.2.1), and the excitation signal has a frequency of 1 MHz, with a varying voltage from 1 Vpp to 4 Vpp. It can be observed that for the circular and square coils, the inductance values change similarly under 1 Vpp, 2 Vpp, 3 Vpp, and 4 Vpp. Thus, the influence of excitation voltage on coil inductance can be neglected.

#### 3.2.4. Effect of Excitation Frequency

Based on the above considerations on the effect of effective area, excitation frequency, and voltage, Position 1, 4 MHz, and 1 Vpp are used to examine the effect of the inductive coil shape on the performance of the eddy current sensor. Figure 13, Figure 14, Figure 15, Figure 16, Figure 17 and Figure 18 show the variation curves of inductance value with measurement distance for the coils of various shapes mentioned in Section 2.2. 

It can be observed that for any coil shape, the inductance value of the coil decreases as the number of turns decreases. Such as, as the number of turns of the circular decreases from 10 to 5, the inductance value of a circular coil at the starting position of 0 mm decreases from 0.63689 μH to 0.37456 μH. Although Types A, D, G, J, M, and P have the same coil turns, the inductance values of Type A and D are significantly higher than those of other types at the same distance, and the variation in inductance values is also much greater compared to those of other shapes. With the same number of turns, circular and square coils have a larger measurement range compared to coils of other shapes, as shown in Figure 16a. The measurement range of Type A and D are 0–7 mm and 0–8 mm, respectively, while the maximum measurement range of Type G, J, M, and P is only 0–6 mm. For circular, square, and vertical rectangular wave coils, their measurement range increases with the number of coil turns. However, the maximum measurement range of coils of other shapes is achieved at turn 7.

Since the variation in turbine blade tip clearance typically ranges from 1–2 mm, the sensitivity and linearity of the coil within the 0–2 mm range are focused. Figure 16b,c shows the sensitivity and linearity of various shapes of sensing coil in the range of 0–2 mm. The circular and square coils have higher sensitivity than the coils of other shapes, as shown in Figure 16b. Type A, D, G, J, M, and P have the same coil turns but their sensitivities are 0.112795 μH/mm, 0.115685 μH/mm, 0.00048 μH/mm, 0.000485 μH/mm, 0.0007 μH/mm, and 0.00088 μH/mm, respectively. It is obvious that the sensitivity of circular and square coils is several times higher than that of coils with other shapes. With the same number of turns, the sensitivity of the square coil is the highest. The linearity of the square coil has also good linearity, as shown in Figure 16c. The linearity of Type D, E, and F is 98.41%, 97.779%, and 98.211%. Moreover, its linearity remains relatively stable with varying numbers of coil turns. Additionally, both circular and rectangular wave coils exhibit relatively high linearity.

In order to investigate the effect of the coil being vertical or horizontal to the blade’s chord direction on the sensor’s inductance characteristics, the comparison of rectangular wave type meander shape coils of Type G, H, and I, and Type J, K, and L is performed. The sensor measurement range, sensitivity, and linearity show some improvement in Type G, H, and I. For example, Type G has a 0–3.5 mm measurement range, 0.000485 μH/mm sensitivity, and 94.528% linearity, but Type J only has a 0–2.5 mm measurement range, 0.000465 μH/mm sensitivity, and 93.645% linearity. This sensitivity improvement is still low compared to other inductive coil shapes, such as circular and square coils. The inductance characteristics do not show a consistent improvement with an increase in the number of coil turns. The comparison has also been made for meander shape inductive coil Type M, N, O, P, Q, and R. The comparisons of sensitivity for both types show similar results. The highest linearity of 98.681% is exhibited by Type J.

## 4. Conclusions

In this paper, we explored various shapes of planar sensing coils for dynamic blade tip clearance measurement, specifically focusing on circular, square, rectangular wave, and triangular wave shapes. To assess the measurement range, sensitivity, and linearity performance, we designed and fabricated various sensing coils and developed an experimental dynamic testing platform. The experimental results indicated that the square coil exhibited superior overall measurement performance, achieving a measurement range of 0 mm to 8 mm, a sensitivity of 0.115685 μH/mm, and a linearity of 98.41% within the 0 mm to 2 mm range. The circular coils also performed well when compared to other meander-shaped coils, such as rectangular wave and triangular wave types. Notably, the circular coil achieved a maximum measurement range of 0 mm to 6.5 mm, a maximum sensitivity of 0.112795 μH/mm, and a linearity slightly below 96% within the range of 0 mm to 2 mm. Although rectangular wave and triangular wave coils offer a high degree of linearity and a wide measurement range, their low sensitivity levels were inadequate for practical applications. Additionally, while increasing the number of coil turns can enhance the measurement range and sensitivity for circular and square coils, this method does not yield significant performance improvements for coils of other shapes. As a result, square coils are more suitable for measuring dynamic blade tip clearance within a range of 0–2 mm. Nonetheless, if the measurement exceeds 5 mm and precise linearity is required, consider using either a rectangular wave coil or a triangular wave coil. Additionally, if space constraints are considerable, the circular coil with an equivalent number of turns may serve as a suitable alternative. Therefore, the appropriate coil shape should be selected based on the specific requirements of the application.

## Figures and Tables

**Figure 1 sensors-24-06133-f001:**
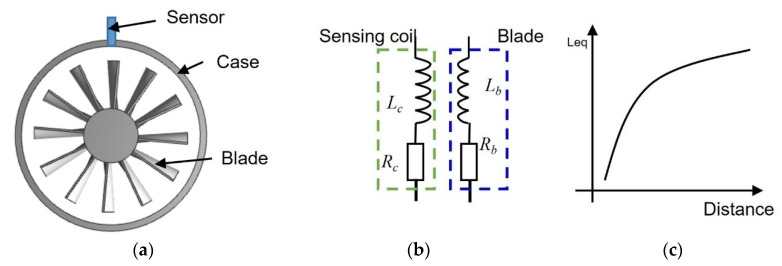
Illustration of the sensing principle of an eddy current sensor for the dynamic tip clearance measurement. (**a**) An eddy current sensor mounted on the turbine case; the sensor is highlighted in blue. (**b**) Equivalent circuit diagram for tip clearance measurement using an eddy current sensor; the sensor is modeled as an inductance (*L_c_*) connected with a resistance (*R_c_*) in series; the blade is modeled in the same way. (**c**) The Lc of the sensing coil is used to measure tip clearance (Distance).

**Figure 2 sensors-24-06133-f002:**
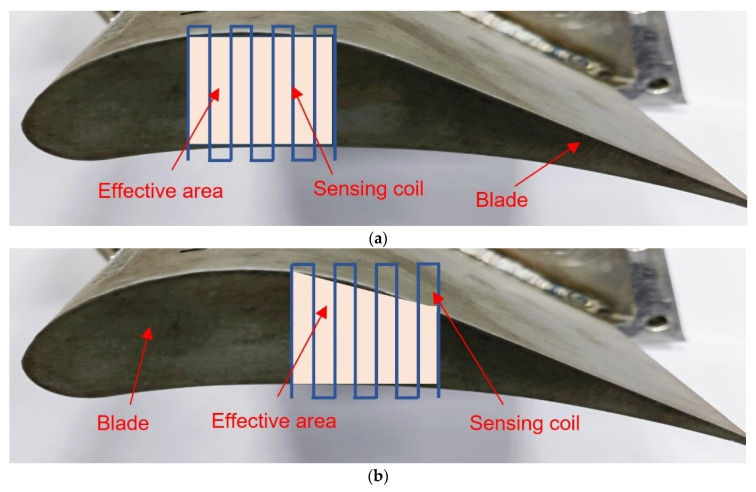
Comparison of inductance effective area of the coil at different blade positions. (**a**) The scheme of the relative position 1 between the sensing coil and the actual blade; (**b**) The scheme of the relative position 2 between the sensing coil and the actual blade.

**Figure 3 sensors-24-06133-f003:**
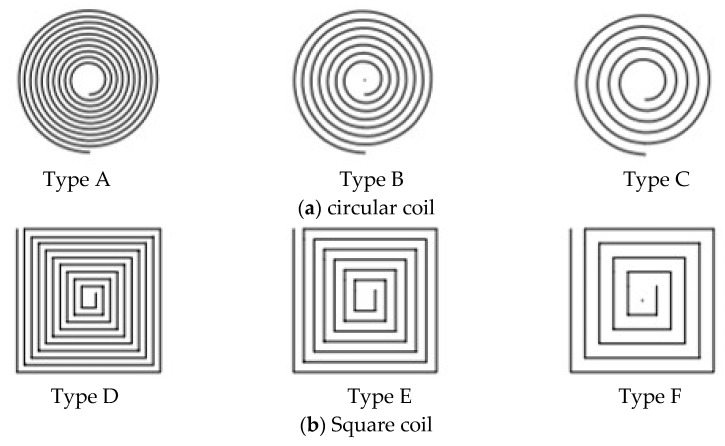
Various sensing coil shapes.

**Figure 4 sensors-24-06133-f004:**
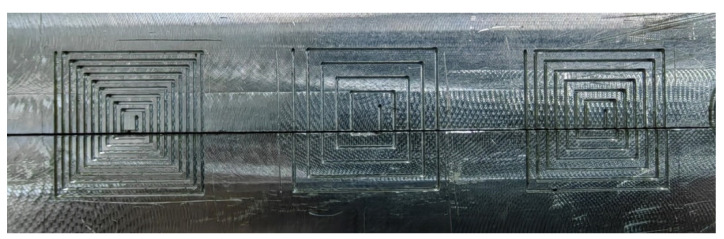
An aluminum mold engraved with partial-shaped coils.

**Figure 5 sensors-24-06133-f005:**
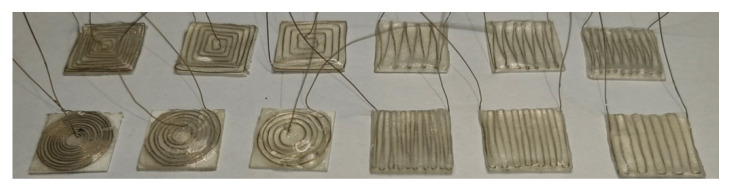
Coils of various shapes.

**Figure 6 sensors-24-06133-f006:**
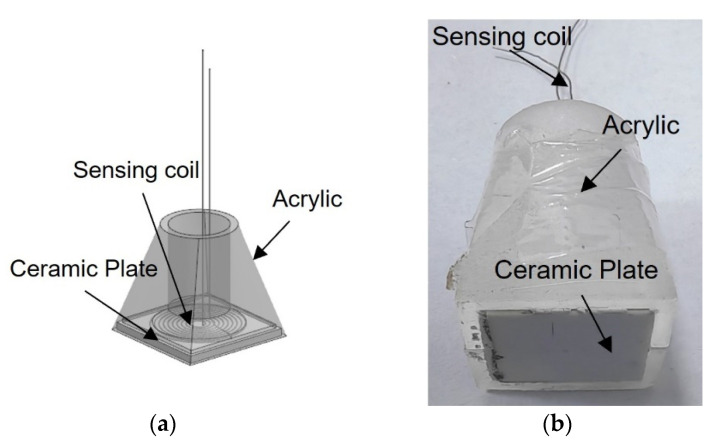
Sensor; (**a**) the design sketch of sensor; (**b**) sensor after encapsulation.

**Figure 7 sensors-24-06133-f007:**
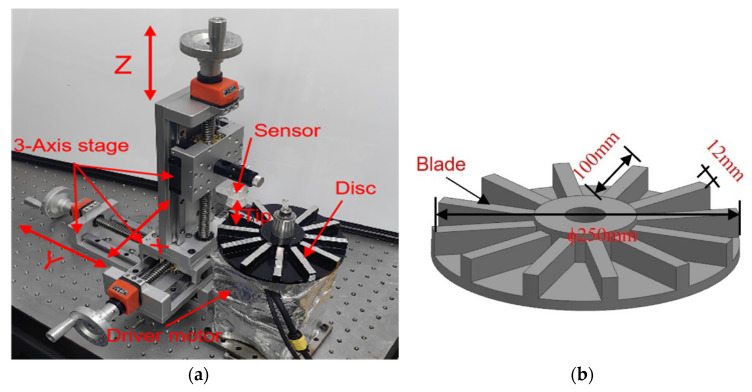
(**a**) Illustration of testing setup for evaluating the sensors with sensing coils of various shapes; (**b**) image of simplified tested disc made of Inconel 718.

**Figure 8 sensors-24-06133-f008:**
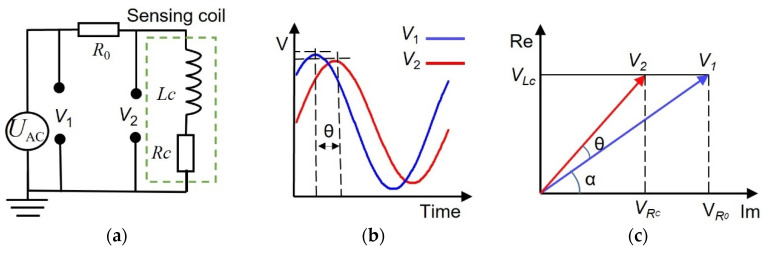
(**a**) The equivalent measurement circuit of the sensing sensor. The sensing coil is modeled as *L_c_* in series with *R_c_*; (**b**) typical output signals of *V*_1_ and *V*_2_; (**c**) The vector diagram of the output signals of *V*_1_ and *V*_2_; (**d**) typical inductive pulses generated by the passage of the blade tips at a rotating speed of 2500 r/min.

**Figure 9 sensors-24-06133-f009:**
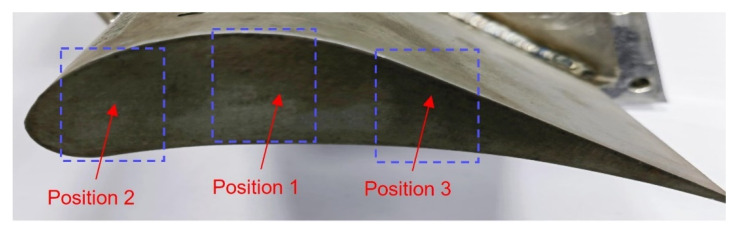
The distribution of three positions along the blade chord.

**Figure 10 sensors-24-06133-f010:**
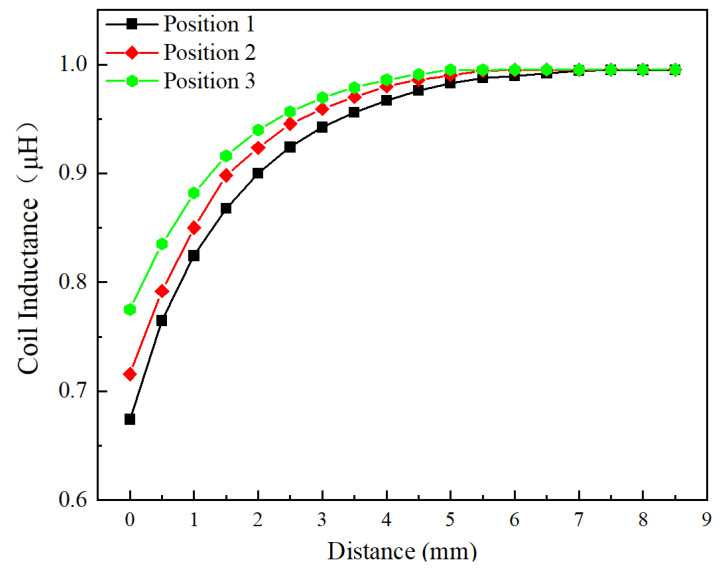
Effect of effective area on the inductance value of the circular shape coil.

**Figure 11 sensors-24-06133-f011:**
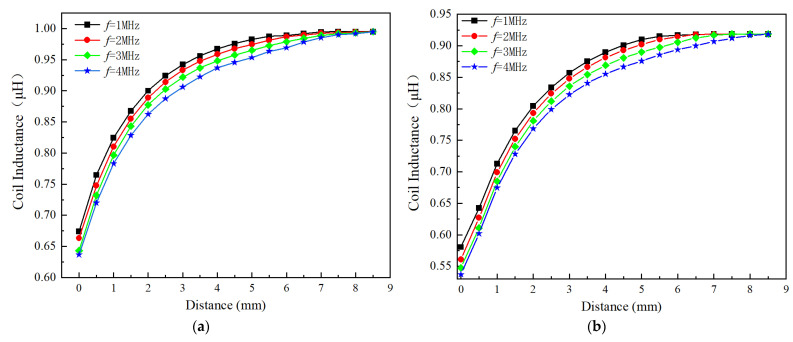
Effect of excitation frequency on the inductance value of the different sensing coil; (**a**) Circular coil; (**b**) Square coil.

**Figure 12 sensors-24-06133-f012:**
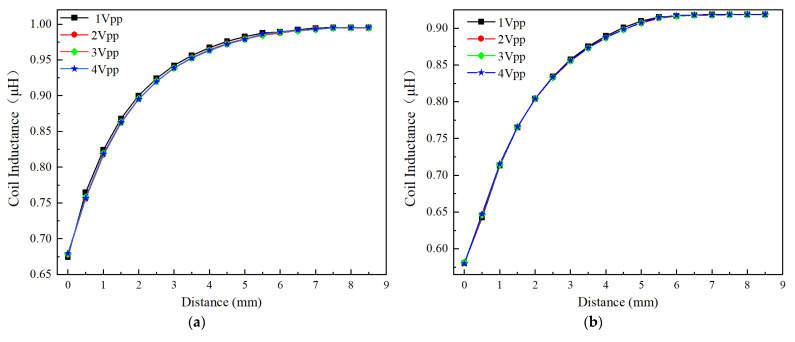
Effect of excitation voltage on the inductance value of the different sensing coil; (**a**) Circular coil; (**b**) Square coil.

**Figure 13 sensors-24-06133-f013:**
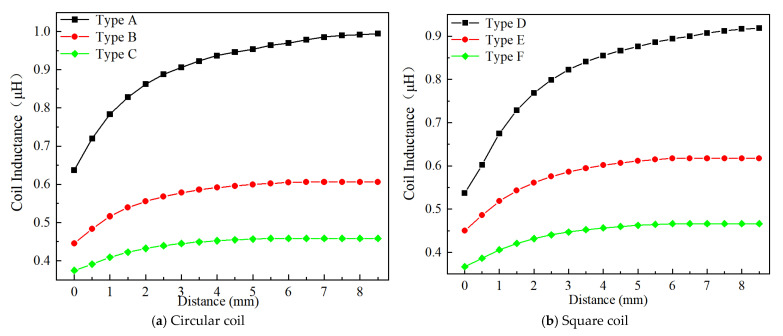
The inductance value of the circular coil.

**Figure 14 sensors-24-06133-f014:**
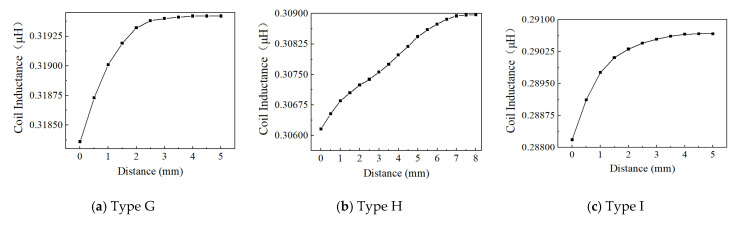
The inductance value of the rectangular wave coil-vertical.

**Figure 15 sensors-24-06133-f015:**
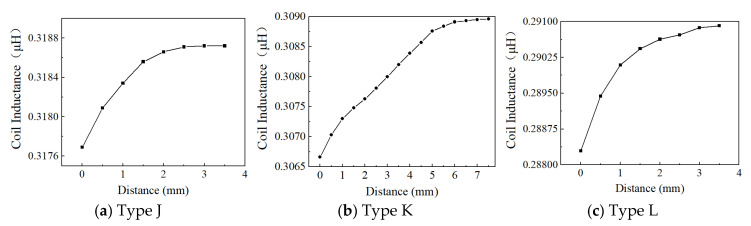
The inductance value of the rectangular wave coil-horizontal.

**Figure 16 sensors-24-06133-f016:**
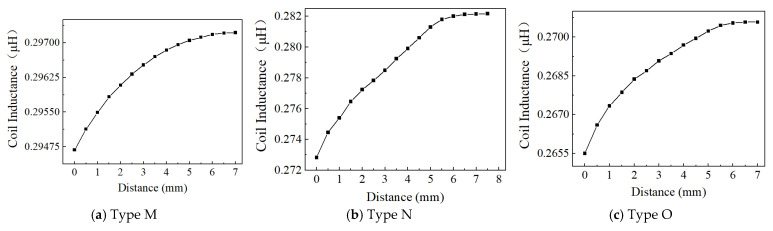
The inductance value of the triangular wave coil-vertical.

**Figure 17 sensors-24-06133-f017:**
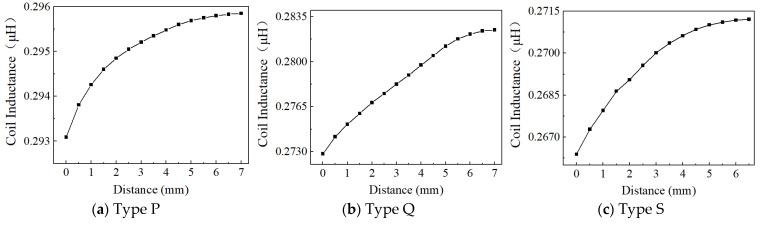
The inductance value of the triangular wave coil-horizontal.

**Figure 18 sensors-24-06133-f018:**
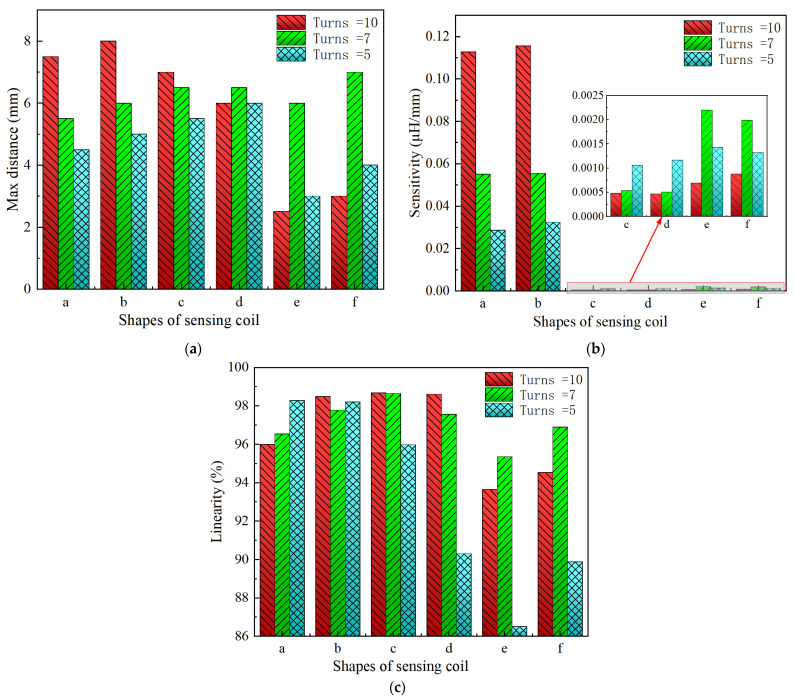
The measurement range, sensitivity, and linearity of various shapes of sensing coil, a–f represent the following coil shapes: circular coil, square coil, rectangular wave coil—vertical, rectangular wave coil—horizontal, triangular wave coil–vertical, triangular wave coil—horizontal, respectively; (**a**) The measurement range of coils with various shapes; (**b**) the sensitivity of coil with various shapes within the range of 0–2 mm; (**c**) the linearity of coil with various shapes within the range of 0–2 mm.

## Data Availability

The data presented in this study are available on request from the corresponding author.

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
