# Peer review of "Experimental Investigation on the Effect of Coil Shape on Planar Eddy Current Sensor Characteristic for Blade Tip Clearance"

_sensors, 2024, doi:10.3390/s24186133_

Round 1

Reviewer 1 Report

Comments and Suggestions for Authors

Dear Authors,

This paper brings an interesting subject about the sensor design to be used in the turbine tip clearance effects on machine performance. Different sensing coil configurations were tested in an adequate rotational speed (until 18k rpm) that represents a real machine rotational speed. 

Some comments to be included in the paper:

1) What is the temperature in which the test was done?

2) Include a real picture of the test facility. You can maintain the illustration in Fig. 7. Show the details of the sendor instalation for both: blade and "casing". The design concept is well described, however include this details.

3) In the case of turbine blade with squealer, is it possible to use this sensor?

4) Explain in a clear way, what can be measure using this sensor technology. For example: is it possible to used it to analyse flow characterists at tip clearance region?

5) Is it possible to use in other axial and/or radial turbomachines?

Include more application of this technology for the turbomachine designer. This will be very interesting.

6) This sensor, can be applied for LPT and HPT?

Kind Regards.

Author Response

1.) The What is the temperature in which the test was done?

Response 1: We are very sorry for our negligence of the detail regarding the experimental temperature. All experiments were conducted at room temperature. We have already added this information in the paper. lines 16 and lines 187

The initial purpose of developing platinum coils was to measure blade tip clearance in high-temperature environment. For specific details, please refer to the references 13-22 in this paper. Since this paper primarily investigates the impact of coil shape on sensor output, experiments in high-temperature environments were not conducted.

2.) Include a real picture of the test facility. You can maintain the illustration in Fig. 7. Show the details of the sensor installation for both: blade and "casing". The design concept is well described, however include these details.

Response 2: Thank you very much for your reminder. We have already added the real picture of the test facility in the paper. Additionally, the article provides details on the sensor installation: the sensors are placed on the horizontally positioned disk to avoid the influence of blade elongation caused by centrifugal force on the sensor output results. It also describes that the tip clearance refers to the distance from the underside of the sensor to the top surface of the blade. Lines: 188-196

3.) In the case of turbine blade with squealer, is it possible to use this sensor?

Response 3: Thank you very much for your question regarding its application scenarios. According to the research of the literature named “The development of a hot section eddy current sensor for turbine tip clearance measurement”, the sensor can be applied to the case of turbine blades with squealers. Since the output performance of the eddy current sensor is related to the cross-sectional area of the target object, the specific application should be tailored to the actual situation.

4.) Explain in a clear way, what can be measure using this sensor technology. For example: is it possible to used it to analyse flow characterists at tip clearance region?

Response 4: Thank you very much for your question. According to the literatures “Axial turbine tip desensitization by injection from a tip trench Part2: Leakage flow sensitivity to injection location”, “Rotor-tip leakage: Part I – Basic methodology” and “Effect of tip clearance and rotor-stator axial gap on the efficiency of a multistage compressor”, it can be seen that excessive tip clearance can not only lead to increased leakage losses and decreased engine efficiency but also make the engine more prone to surging. It means that the magnitude of leakage can be analyzed by measuring tip clearance. And it can be used to predict the engine's efficiency and the likelihood of surge. The important thing is it can be used to analyze whether the blade has rubbed against the casing, thereby preventing accidents.

Additionally, it can be used for blade tip timing (BTT) based on the literature “The Use of Eddy Current Sensor for the Measurement of Rotor Blade Tip Timing Development of a New Method Based on Integration”. BTT can monitoring blades conditions so that dangerous conditions can be detected in a timely manner.

In summary, eddy current sensors can be used for health monitoring of aircraft engines.

5.) Is it possible to use in other axial and/or radial turbomachines?

Include more application of this technology for the turbomachine designer. This will be very interesting.

Response 5: Thank you very much for your questions regarding its application scenarios. Eddy current sensors can indeed be applied to axial and/or radial turbomachines, and their use is not limited to blade tip clearance measurements. They can also be employed for temperature detection as well as vibration monitoring in both axial and radial turbomachines. Such as the literature “The development of a hot section eddy current sensor for turbine tip clearance measurement” has carried out exploration on the application of eddy current sensors in evaluation of turbine axial movement. The literature “experimental investigation of high temperature-resistant inductive sensor for blade tip clearance measurement” has explored the potential of eddy current sensors in temperature measurement and has confirmed that eddy current sensing can be applied to temperature measurement.

Overall, the application of eddy current sensors in turbomachines can enhance performance monitoring and maintenance efficiency. They provide designers with more data and information, helping them optimize the design and operation of turbomachines.

6.) This sensor, can be applied for LPT and HPT?

Response 6: Thank you very much for your questions. Based on the literature “Tip clearance measurements on an engine high pressure turbine using an eddy current sensor”, the eddy current sensor with platinum group coil can be able to operate at high temperatures of about 1400℃. And the experiment of tip clearance measurements in the high-pressure turbine stage of a jet engine has been successfully conducted. It means that this sensor can also applied for blade tip clearance measurement in LPT.

Reviewer 2 Report

Comments and Suggestions for Authors

In order to improve the measurement accuracy of eddy current sensors for the Blade Tip Clearance,the experimental study on the influence of eddy current coil shape on the measurement accuracy of Blade Tip Clearance was conducted in the paper. The studied coil shapes include circles, squares, rectangular waves, and triangular waves. And an optimal coil shape to enhance sensing performance was proposed. The research results of the paper can provide guidance for selecting coil shapes and optimizing the performance of planar eddy current sensors. The opinions and suggestions on the paper are as follows:(1)In Figure 5, the geometric parameter errors of the handmade sensor is relatively large, will these errors greatly affect the accuracy of the test results?  (2)The English expression of the paper needs further optimization. (3)At the end of the paper,  the following conclusion is drawn: " Therefore, square coils are more suitable for measuring blade tip clearance. However, the appropriate coil shape should be selected based on the specific requirements of the application”.  For the specific application scenarios described in the paper, clear recommendations for selecting sensor shapes should be provided at the end. (4)"Each type of sensing coil has the same external dimensions (10mm x 10mm)" and "In Figure 7, the width of the tested disc made of Inconel is 12mm". When the sensor is smaller than the tested object, circular sensors exhibit good performance in many situations due to their symmetry. Please provide further reasons for the final recommendation of choosing square sensors in the paper.

Comments on the Quality of English Language

The English expression of the paper needs further optimization.

Author Response

  • In Figure 5, the geometric parameter errors of the handmade sensor is relatively large, will these errors greatly affect the accuracy of the test results?

 Response: Thank you very much for your concern about the the geometric parameter errors of the handmade coils. Currently, the manufacturing processes for single-layer planar platinum coils mainly include LTCC and HTCC. However, these two processes for planar platinum coils are still not fully mature and are still in the experimental research stage. Therefore, manual winding is the only option, similar to the methods used in the literature for sensor fabrication, named “Experimental investigation of high temperature-resistant inductive sensor for blade tip clearance measurement”. At the beginning of this study, this issue was taken into consideration. Two approaches were employed to minimize production errors in the sensors. First, a coil fabrication template was designed, as shown in Figure 4, to reduce the unevenness of the wiring during manual winding. Second, for each coil shape, three different coil turns were produced to compare whether the inductance of the coils increased with the number of turns. Figure 15 shows that regardless of the coil shape, the inductance showed the same trend with changes in the number of turns. This indirectly proves that the manual fabrication method did not significantly affect the results.

2.) In The English expression of the paper needs further optimization. 

 Response: Thank you very much for this valuable suggestion, and the whole manuscript has been polished accordingly.

3.)At the end of the paper,  the following conclusion is drawn: " Therefore, square coils are more suitable for measuring blade tip clearance. However, the appropriate coil shape should be selected based on the specific requirements of the application”.  For the specific application scenarios described in the paper, clear recommendations for selecting sensor shapes should be provided at the end.

Response: Thank you very much for your suggestion. We have refined the expression to make it more coherent. Recommendations regarding the application ranges of different coils have been added. Such as if the measurement exceeds 5mm and precise linearity is required in your coil design, consider using either a rectangular wave coil or a triangular wave coil. If space constraints are significant, the circular coil with the same number of turns compared to the other shapes might be suitable. Lines 355-360.

4.)"Each type of sensing coil has the same external dimensions (10mm x 10mm)" and "In Figure 7, the width of the tested disc made of Inconel is 12mm". When the sensor is smaller than the tested object, circular sensors exhibit good performance in many situations due to their symmetry. Please provide further reasons for the final recommendation of choosing square sensors in the paper.

Response: Thank you very much for your question. We apologize for the fact that this paper does not investigate the impact of sensor shapes on the output results for that the sensors are smaller compared to the size of the target object. There are two main reasons why we did not conduct research on smaller-sized coils. The main reason is the limitations of the processing technology for platinum coils. Additionally, incorporating small-sized coils of different shapes would make the manufacturing process even more difficult. Therefore, considering practical conditions, we handcrafted coils of different shapes with external dimensions of 10mm x10mm.

And based on the results of this paper, the article focuses on the performance of the sensor from three aspects: linearity, sensitivity, and measurement range. The square coils perform better than circular coils in terms of measurement range and linearity of sensitivity within the 0-2mm range. And it can be observed that circular sensors and square sensors exhibit similar symmetry in distribution of magnetic flux density in the literature named “Analytical study of magnetic flux density for circular spiral and square spiral coils”. And the literature “Magnetic Field Calculation and Simulation Analysis of Close-Range Magnetic Induction Coil” states that under the same conditions, the magnetic induction strength at the center of the square coil is 1.01 times that of the circular coil. Therefore, combining the experimental results and the mentioned literature, square coils exhibit better measurement performance compared to circular coils under the same conditions.

However, the impact of sensor shapes on the output results for small-sized sensors which are smaller compared to the size of the target object should be studied, and why circular coils are more frequently used for measuring small targets is also worth investigating. These are also the content we need to investigate next.
